# Determining the patency of biliary tracts in dogs with gallbladder mucocele using near-infrared cholangiography with indocyanine green

**Su-Hyeon Kim**[1,2], **Sungin Lee**[1]*

1 Department of Veterinary Surgery, College of Veterinary Medicine, Chungbuk National University, Cheongju, Chungbuk, South Korea, 2 Department of Veterinary Surgery, Heamaru Referral Hospital, Seongnam, Korea

* sunginlee@cbnu.ac.kr

**Data Availability Statement:** All relevant data are within the manuscript and its Supporting Information files.

## Abstract

Cholecystectomy is indicated for gallbladder mucoceles (GBM). Evaluating the patency of the biliary duct and precise biliary tree visualization is crucial for reducing the risk of compromised bile flow after surgery. Therefore, intraoperative cholangiography (IOC) is recommended during cholecystectomy to prevent biliary tract injury. Although indocyanine green (ICG) cholangiography has been extensively reported in human medicine, only one study has been conducted in veterinary medicine. Therefore, this study aimed to demonstrate the use of ICG for IOC to identify fluorescent biliary tract images and determine the patency of the common bile duct during cholecystectomy in dogs. This study comprised 27 dogs, consisting of 17 with gallbladder mucoceles (GBM) and 10 controls, specifically including dogs that had undergone elective cholecystectomy for GBM. ICG injection (0.25 mg/kg) was administered intravenously at least 45 minutes before surgery. During the operation, fluorescent images from cholangiography were displayed on the monitor and obtained in black-and-white mode for the comparison of fluorescence intensity (FI). The FI values of the gallbladders (GBs) and common bile duct (CBD) were measured using FI analyzing software (MGViewer V1.1.1, MetapleBio Inc.). The results demonstrated successful CBD patency identification in all cases. Mobile GBM showed partial gallbladder visibility, whereas immobile GBM showed limited visibility. Additionally, insights into the adequate visualization of the remaining extrahepatic biliary tree anatomy were provided, extending beyond the assessment of CBD patency and gallbladder intensity. Our study demonstrates the potential of fluorescent IOC using intravenous injection of ICG for assessing the patency of the cystic duct and common bile duct during cholecystectomy in patients with GBM, eliminating the need for surgical catheterization and flushing of the biliary ducts. Further research is warranted to investigate and validate the broader applicability of ICG cholangiography in veterinary medicine.

**Funding:** This work was supported by the National Research Foundation of Korea (NRF) grant funded by the Korea government (MSIT) (RS-2023-00253736) and also supported by the Basic Research Lab Program (2022R1A4A1025557) through the National Research Foundation (NRF) of Korea, funded by the Ministry of Science and ICT.

**Competing interests:** The authors have declared that no competing interests exist.

## Introduction

Gallbladder mucocele (GBM), characterized by the accumulation of excess mucin and bile in the gallbladder (GB), can cause partial or complete extrahepatic biliary obstruction (EHBO) by extending bile-laden mucus into the cystic, hepatic, and common bile ducts (CBDs) [1]. As a result of biliary tract outflow obstruction and progressive GB distension, pressure necrosis of the GB or bile duct may result in rupture and subsequent bile peritonitis, which can be life-threatening. Cholecystectomy is indicated for GBM treatment [2]. It is crucial to confirm the patency of the CBD and obtain adequate visualization of the anatomy of the extrahepatic biliary tree to ensure safe dissection and reduce the risk of compromised bile flow after surgery resulting from residual bile plugs or gallstones. Generally, catheterization and flushing of the biliary ducts, accomplished by normograde or retrograde techniques, are proposed to ensure patency [3]. However, intraoperative catheterization remains controversial because of the risk of iatrogenic damage to the biliary tract and prolonged surgical time [4, 5].

Initially introduced by Mirizzi in 1937, intraoperative cholangiography (IOC) is recommended during cholecystectomy to prevent biliary tract injuries [6]. In the field of human medicine, intraoperative near-infrared (NIR) fluorescent cholangiography using a real-time imaging technique with intravenous administration of indocyanine green (ICG) has been widely used to assess fluorescent images of the biliary tract, enabling real-time assessment of fluorescent images of the biliary tract, and reducing biliary tract injury [7]. NIR fluorescence with a wavelength of 750–900 nm is not visible to the human eye [8]. Its novel, less invasive, and non-radioactive technique can enable the real-time assessment of fluorescent images of the biliary tract without altering the surgical field. Using IOC, surgeons can obtain accurate information on bile duct patency. Establishing IOC as a routine procedure may encourage surgeons to make better prognoses during cholecystectomy.

Although ICG cholangiography has been extensively reported in human medicine, only one study that conducted cholangiography in normal dogs has been reported in veterinary literature [9]. On the other hand, in this study, we attempted to investigate the cases of dogs with GBM. We hypothesized that IV injection of ICG could provide fluorescent images of the biliary tract and determine the patency of the CBD without necessitating surgical catheterization or flushing of the ducts. The objective of this study was to describe a technique for performing fluorescent IOC using IV injection of ICG to provide fluorescent images of the biliary tracts and to determine the patency of the CBD in client-owned dogs with normal GBs and those with GBM.

## Materials and methods

### Study population and preoperative assessment

Client-owned dogs were prospectively recruited for the study from February 2021 until August 2023. The subjects comprised 27 client-owned dogs with GBM who underwent cholecystectomy (n = 17). A total of 10 control patients with normal GBs undergoing open abdominal surgery were also included. All GBs in the control group were determined to be normal based on abdominal ultrasound and blood analysis.

Dogs with GBM were diagnosed using abdominal ultrasound at the Department of Veterinary Radiology, College of Veterinary Medicine, Chungbuk National University. GBM was confirmed by histopathological assessment after surgery. Only dogs that had undergone elective cholecystectomy for GBM were included in the study. Dogs that were defined as non-elective, if there was evidence of clinical signs attributed to hepatobiliary disease with associated hyperbilirubinemia or if there was evidence of ultrasonographic findings associated with

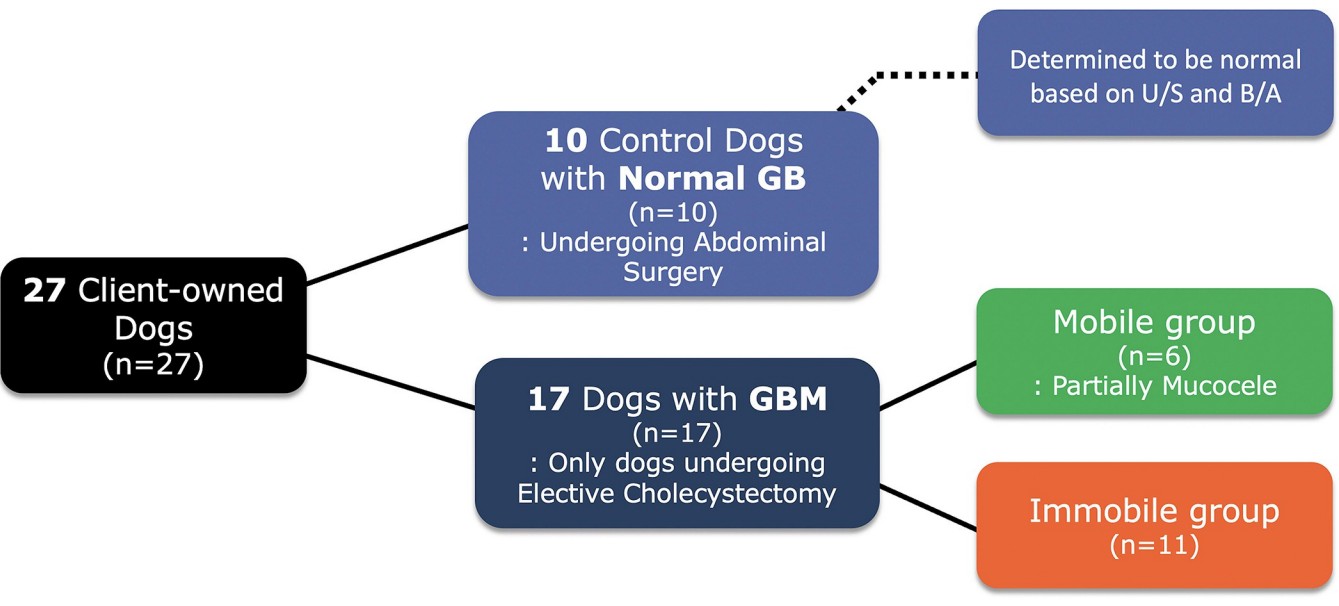

**Fig 1. Patient enrollment.**

reactive changes, impending rupture, rupture of the GBM, or physical obstruction of the CBD were excluded from this study. Dogs with none of the above criteria or an incidental finding of GBM were deemed elective and enrolled in the study. The 17 dogs with GBM that participated in this study were divided into two groups depending on the mucocele stage: an immobile GBM group (n = 11) and a mobile GBM group (n = 6) (Fig 1). The GBs were classified into three groups: (i) normal gallbladder; (ii) mobile GBM; and (iii) immobile GBM. The diagnosis of GBM depends on imaging studies, blood test results, and clinical signs, such as anorexia, vomiting, and lethargy [10]. Abdominal ultrasonography is one of the most useful diagnostic tools for evaluating GBM [11]. On abdominal ultrasonography, immobile and echogenic bile with a finely striated or stellated pattern is typically observed within the GBM [12]. This distinguishes it from GB sludge, which exhibits bile movement independent of gravity [13].

All dogs underwent physical and imaging examinations, including radiography and abdominal ultrasonography, complete blood counts, and serum biochemical analysis. The biochemical panel included ALP, ALT, GGT, AST, cholesterol, bilirubin, and triglyceride levels, which are associated with hepatobiliary disease. All the samples were analyzed using a Hitachi 7180 automated biochemical analyzer (Hitachi, Tokyo, Japan) equipped with commercial kits (JW Pharmaceutical, Seoul, Republic of Korea).

### Informed consent

All patients were client-owned dogs that underwent abdominal surgery at the Department of Veterinary Surgery, Veterinary Medical Teaching Hospital of Chungbuk National University. Written informed consent was obtained from each owner, and all procedures were approved by the Chungbuk National University Institutional Animal Care and Use Committee (CBNUA-2190-23-02).

### Administration of ICG

As a source of fluorescence, 0.25 mg/kg ICG (25 mg of ICG, Cellbion Green, Cellbion Co., Ltd., Seoul, South Korea) was injected intravenously at least 45 min before surgery, up to a

maximum of 2 hours. About 25 mg vial was dissolved in 10 mL of sterile water, and 1 mL of the dissolved solution was further diluted with an additional 9 mL of sterile water to create a 0.25/mL solution immediately before injection. ICG was administered only to patients without renal failure or a history of adverse reactions to the iodine solution.

### Fluorescent imaging system

Intraoperative ICG display was performed using an intraoperative NIR fluorescence-guided imaging system (Metaple Bio, Seoul, South Korea). It comprises a control unit and a 'NIR' light source. The NIR light source produces a wavelength of 802.5 nm, and this imaging system detects infrared light above the bile duct. The distance between the laser output and surgical site, which was the working distance in the fluorescence imaging system, was 40 cm. During the operation, fluorescent images of the cholangiography were displayed on the monitor and obtained in black-and-white mode to compare the fluorescence intensity (FI). FI values of the GBs and CBD were measured using FI analyzing software (MGViewer V1.1.1, MetapleBio Inc.) (Fig 2). Bile duct patency was determined by comparing FI values. In the control group, values for the gallbladder (GB) and common bile duct (CBD) were established. When FI values of the CBD in GBM patients closely resembled those in the control group, CBD patency was considered confirmed. This criterion ensured a thorough evaluation of bile duct patency while minimizing unnecessary interventions. Cholecystectomy was only performed for GBM dogs and those with GB sludge who underwent liver lobectomy for visualization.

### Monitoring

After administration of ICG, the patients were closely monitored for any signs of adverse reactions involving the cardiac, or respiratory (anaphylaxis) [14]. Monitoring included continuous observation for changes in heart rate, respiratory rate, mean arterial pressure, and oxygen saturation during the anesthesia. After the patients recovered from the anesthesia, close monitoring of physical examination, heart rate, respiratory rate, mean arterial pressure and oxygen saturation were continuously conducted using monitoring devices. Additionally, blood analysis and hepatic ultrasonography were performed within one week following surgery, with

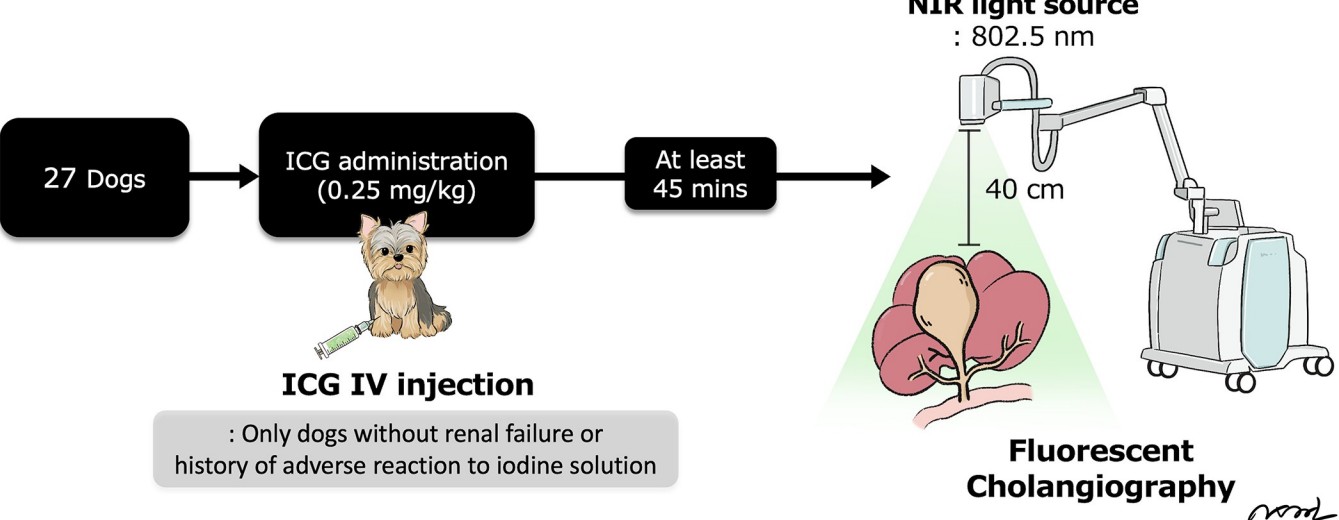

**Fig 2. Flowchart of the near-infrared fluorescent cholangiography using indocyanine green (ICG) study procedure © 2023 by Sungin Lee is licensed under CC BY 4.0.**

follow-up examinations conducted one week, two weeks, and one month postoperatively to assess for any delayed adverse effects or complications.

## Statistical analysis

Statistical analyses were performed using the SPSS statistical software package, version 24.0. (SPSS Inc., Chicago, IL, USA). The Shapiro–Wilk method was used to evaluate whether the obtained data were normally distributed. An independent t-test was used to analyze the differences in age and total cholesterol between the control group with normal GB and the GBM group. The Mann–Whitney U-test was performed to analyze differences in body weight and biochemical characteristics, such as triglyceride, ALT, AST, ALP, GGT, and total bilirubin levels, between the control group with normal GB and the GBM group. Data were presented as means and standard deviations. Welch's ANOVA was performed to analyze the differences in FI among the following groups: controls (Normal GB), dogs with immobile GBM, and dogs with mobile GBM. If a significant difference was detected, the Games–Howell test was performed for post-hoc comparisons. Data were presented as means and standard deviations. Statistical significance was set at P-value<0.05.

## Results

### Patients' characteristics

Basic information on the 10 dogs with normal GBs (n = 10) and 17 dogs with GBM (n = 17) are summarized in Table 1. The median age of the dogs with GBM was 12 years, which was higher than that of control dogs with normal GB (9 years old) ($p = 0.002$). Total cholesterol, triglyceride, and ALP levels were higher in dogs with GBM than in dogs with normal GB ($p = 0.042$, $p = 0.003$, and $p = 0.007$, respectively). Body weight and ALT, AST, GGT, and total bilirubin levels did not significantly differ between the groups.

### IOC using ICG fluorescence imaging systems

The results of fluorescence imaging are presented in Fig 3. The results of fluorescence imaging are presented in Fig 3. The fluorescent cholangiography using ICG identified the CBD in all 27 patients (n = 27). In dogs with normal GBs (n = 10), intraoperative fluorescent cholangiography using a NIR fluorescent imaging system also detected the GB, cystic ducts, and hepatic ducts. The entire length of the cystic and CBDs could be visualized, as well as the junction of the cystic and CBDs (Fig 3A). Interestingly, the GB was partially visualized in dogs with mobile GBM (n = 6). The part where the mucocele was attached to the ventral side of the GB was not detected using the NIR fluorescence camera, whereas the other part was visualized (Fig 3B). Mucoceles were not detected on ICG fluorescence cholangiography (Fig 3B and 3C). In dogs with immobile GBM (n = 11), the entire GB could not be visualized with ICG fluorescence. The entire length of the cystic duct was also not visualized in a group of immobile GBM (Fig 3C). Bile duct patency was determined by comparing FI values. FI values of CBD in the GBM group showed similar FI values in the control group, there, no flushing was performed in GBM groups in which CBD patency was confirmed through FI value comparison. Fluorescence of the CBD persisted until the closure of the abdomen in both the control and the GBM groups. Moreover, ICG cholangiography revealed the passage into the duodenum in both the control and the GBM groups (Fig 4). ICG fluorescence was also readily visible in the liver.

All patients successfully recovered from the anesthesia. None of the dogs showed adverse reactions to ICG. Postoperative follow-up was conducted on all patients, including those with immobile GBM, for several weeks after surgery. During these follow-up periods, clinical assessments,

**Table 1. Comparison of age, sex, breed, body weight (BW), and biochemical characteristics between dogs with normal gallbladder (controls) and dogs with gallbladder mucocele.**

| | Dogs with normal GB (n = 10) | Dogs with GBM (n = 17) | P |
|---|---|---|---|
| Age (years) | 9.00 (2.25) | 12.00 (2.00) | 0.002 |
| Sex (n) | Entire female (4) | Entire female (8) | |
| | Entire male (2) | Entire male (2) | |
| | Spayed female (4) | Spayed female (6) | |
| | Castrated male (0) | Castrated male (1) | |
| Breed (n) | Poodle (3) | Maltese (4) | |
| | Maltese (2) | Cocker-Spaniel (3) | |
| | Mixed breed (1) | Poodle (2) | |
| | Shih-Tzu (1) | Shih-Tzu (2) | |
| | Pomeranian (1) | Spitz (2) | |
| | Chihuahua (1) | Pomeranian (1) | |
| | Cocker-Spaniel (1) | Chihuahua (1) | |
| | | Golden Retriever (1) | |
| | | Yorkshire Terrier (1) | |
| BW (kg) | 5.41 (3.60) | 5.17 (5.90) | 0.530 |
| Total cholesterol (mg/dL) | 194.5 (68.75) | 234 (109) * | 0.042 |
| Triglycerides (mg/dL) | 53 (16.75) | 121 (49.00) * | 0.003 |
| ALT (IU/L) | 43 (18.00) | 79 (83.00) | 0.063 |
| AST (IU/L) | 22 (14.25) | 23 (12.00) | 0.724 |
| ALP (IU/L) | 32 (26.25) | 379 (528.00) * | 0.007 |
| GGT (IU/L) | 5 (2.75) | 6 (10.00) | 0.173 |
| T.bil (mg/dL) | 0.025 (0.03) | 0.02 (0.05) | 0.818 |

Data are expressed as the median with the interquartile range (presented in brackets).

**n**, Number of patients; **BW**, bodyweight; **ALT**, alanine transaminase; **AST**, aspartate transaminase; **ALP**, alkaline phosphatase; **GGT**, gamma-glutamyl transferase; **T.bil**, Total bilirubin.

Reference ranges are as follows: Total cholesterol [112–312]; triglycerides [21–133]; ALT [21–102]; AST [23–66]; ALP [29–97]; GGT [1–10]; and T.bil [0.1–0.5].

*p<0.05

ultrasonography, and blood tests were performed for monitoring. All patients showed favorable outcomes, with no complications observed up to a 6-month follow-up period.

### Analysis of FI in control dogs with normal GB, dogs with mobile GBM, and dogs with immobile GBM

Dogs with GBM were divided into two groups: dogs with mobile GBM and those with immobile GBM. The FI of the GB in dogs with a normal GB (Average 15569-pixel values) was significantly higher than that of dogs with GBM of both mobile (Average 4494-pixel values) and immobile (Average 735-pixel values) (P<0.001 and P<0.001, respectively). Furthermore, in dogs with GBM, the FI was higher in the mobile GBM groups (Average 4494-pixel values) than in the immobile group (Average 735-pixel values) (P<0.001) (Fig 5). However, the FI of the CBD did not significantly differ between the group with normal GB and groups with GBM (P = 0.079) (Fig 6).

## Discussion

In this study, NIR cholangiography was successfully performed in all 27 client-owned dogs, revealing the patency of the CBD with minimal invasiveness. The CBD was visualized using an

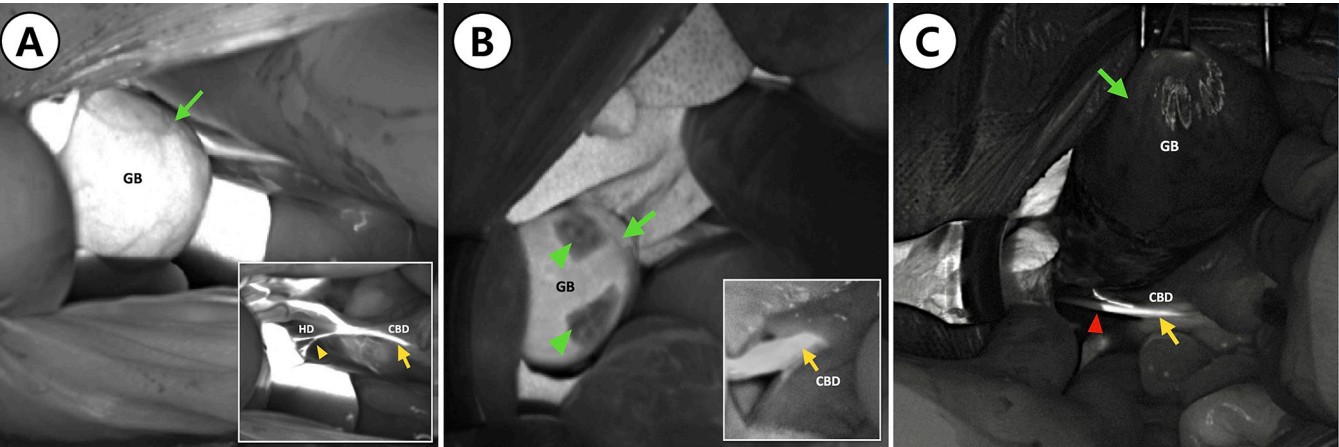

**Fig 3. Fluorescent intraoperative cholangiography.** (A) In a dog with a normal gall bladder, indocyanine green (ICG) fluorescence of the gallbladder (green arrow), the common bile duct (yellow arrow), and the hepatic ducts (yellow arrowheads) are shown. (B) In a dog with a mobile gallbladder, ICG fluorescence of the gallbladder (green arrow) and the common bile duct (yellow arrow). Some parts of the gallbladder (green arrowheads) did not fluoresce because the mucocele was not detected under NIR light source. (C) In a dog with an immobile gallbladder, ICG fluorescence of the gallbladder (green arrow) and the common bile duct (yellow arrow) are shown. Some parts of the cystic duct did not fluoresce (red arrowhead).

NIR camera in all patients until the end of the surgery. Furthermore, NIR cholangiography using ICG made it possible to delineate the structure of the biliary trees, including the passage of fluorescence into the duodenum. Fluorescent images revealed filling defects in the GBs of 17 dogs with GBM. We believe that extrahepatic biliary obstruction may be demonstrated using this technique.

Hyperlipidemia, characterized by elevated serum levels of total cholesterol, triglycerides, or both, is related to the formation of GBM [15]. In the present study, blood concentrations of total cholesterol and triglycerides were higher in dogs in the GBM group than in controls with healthy GBs. 11 dogs with immobile GBM showed striated, stellate, or mixed patterns, and 6 dogs with mobile GBM partially showed a striated or stellate pattern with biliary sludge. Elective cholecystectomy is strongly recommended for patients with GBM. Studies have consistently demonstrated that dogs undergoing elective surgery exhibit lower mortality rates compared to those undergoing nonelective procedures. Performing elective cholecystectomy before the onset of clinical signs significantly increases the chances of a successful outcome [2]. In veterinary medicine, there has been ongoing discussion regarding the confirmation of biliary tract patency even in elective surgeries [16]. Therefore, IOC has been regarded as a potential technique enabling the assessment of patency and visualization of biliary trees [6].

IOC has been recommended to assess biliary trees in humans [17]. Bile duct injury due to misinterpretation of biliary anatomy constitutes a serious complication of cholecystectomy [18]. Over the past few years, fluorescence-guided surgery using ICG during cholecystectomy has emerged as a new technology that allows for real-time enhanced visualization and identification of extrahepatic biliary structures without the use of radiation [19]. During IV injection, ICG rapidly binds to intravascular plasma proteins, which are then cleared by the liver, specifically taken up by hepatocytes, and excreted into the bile. Therefore, protein bound ICG by NIR light causes fluorescence and visualizes the biliary trees [20]. As a result, ICG fluorescence cholangiography facilitates the accurate identification of biliary tracts and high-risk areas that should be monitored before resection and allows for the localization of important landmarks. One of the advantages of fluorescent IOC is its ability to provide a reliable roadmap of the bile duct anatomy and identify the CBD before dissection. This can reduce the chances of

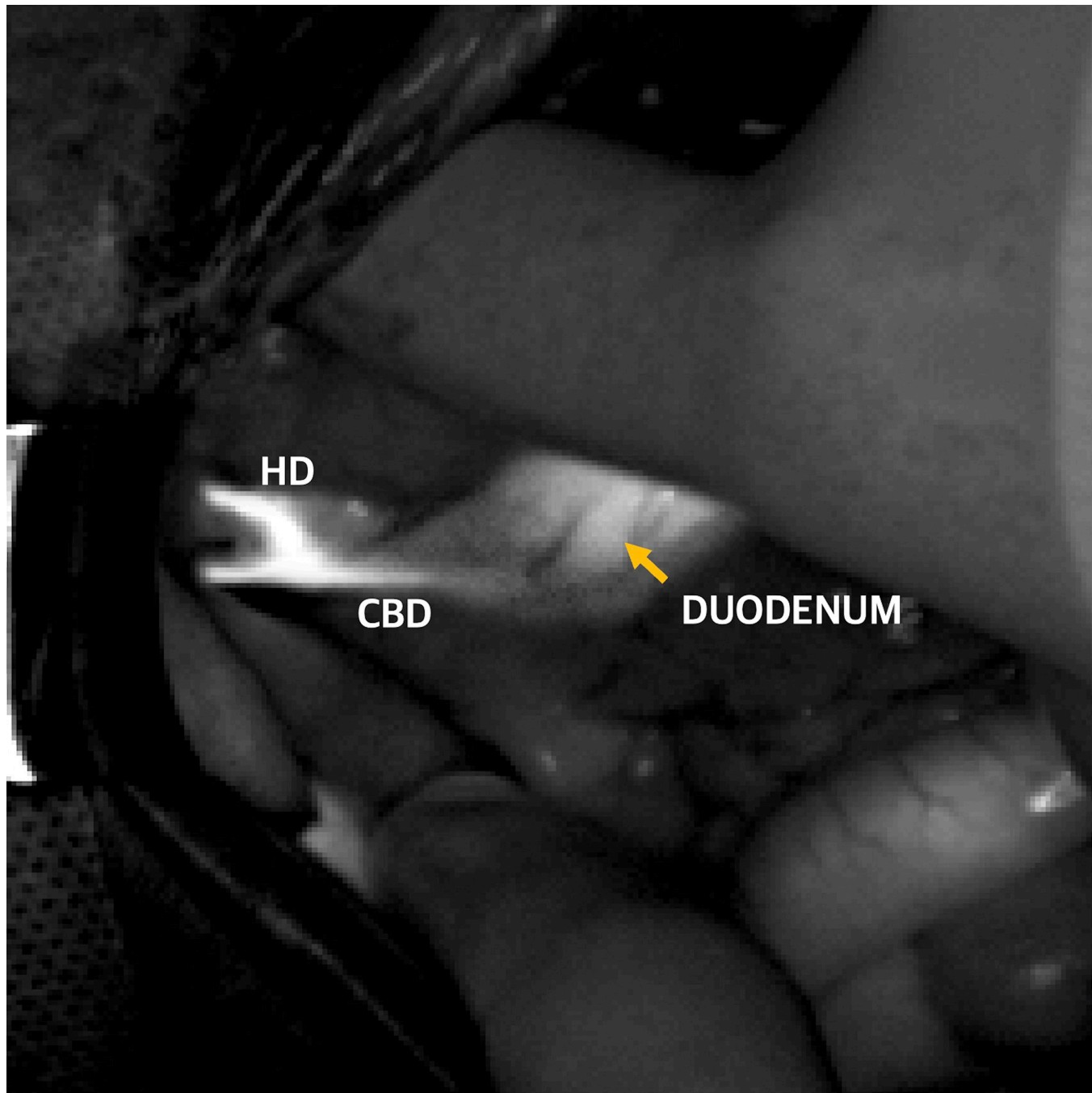

**Fig 4. Intraoperative view using near-infrared camera after ICG administration.** Image of intraoperative ICG cholangiography in a dog. The yellow arrow indicates ICG observed in the passage into the duodenum.

iatrogenic injury and assist surgeons in selecting the best location for dissection [21]. In addition, with fluorescent cholangiography, the cystic duct can be identified without the need to dissect Calot's triangle or insert a transcystic tube for the injection of contrast material, procedures which have a risk of causing bile duct damage [22]. Finally, compared with radiographic cholangiography, fluorescent cholangiography with IV injection of ICG offers safety and is

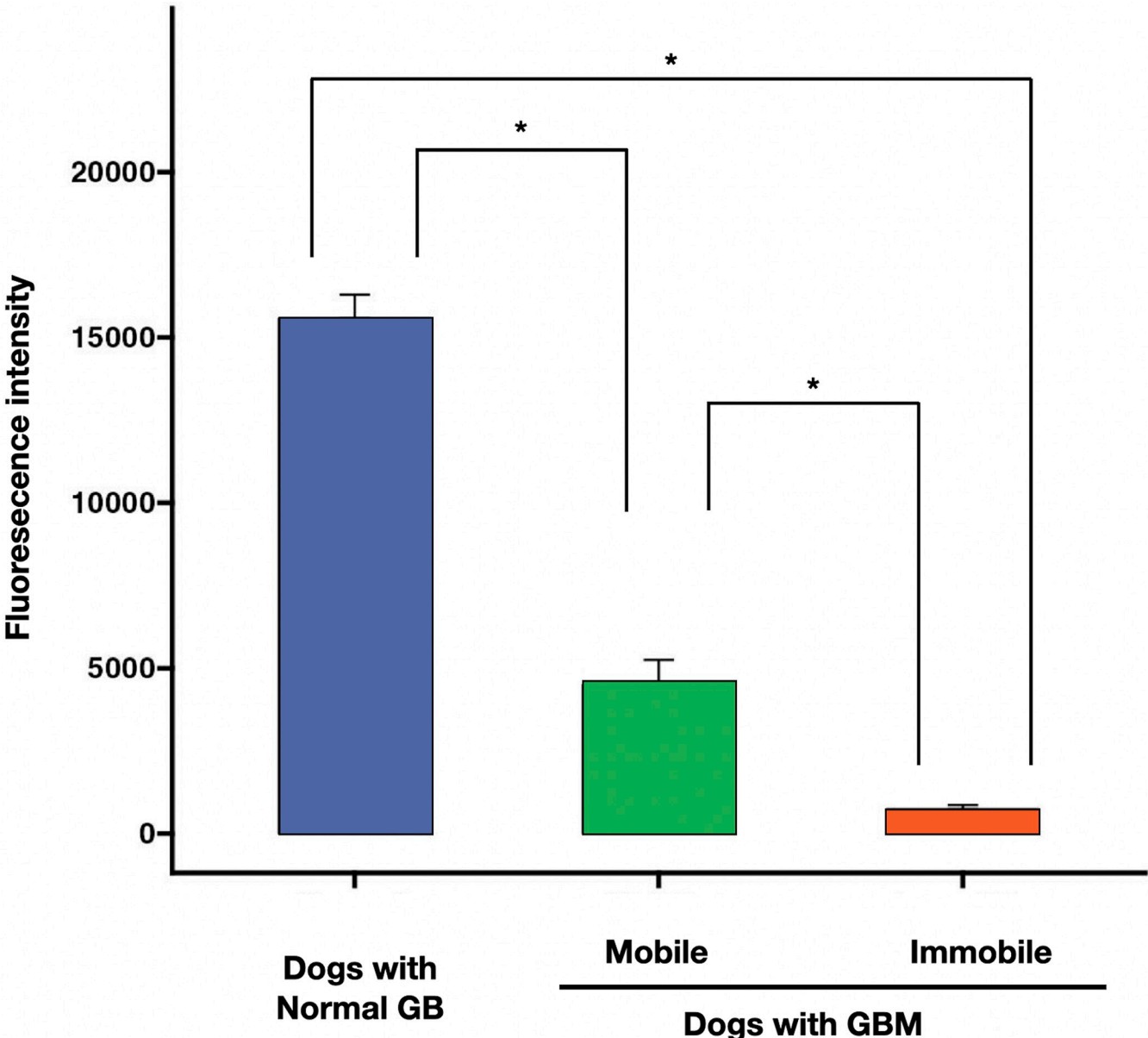

**Fig 5. Fluorescent intensity of the gallbladder (GB).** Comparison of the FI of the gallbladder (GB) among the following groups: controls with normal GB (n = 10), dogs with mobile GBM (n = 6), and dogs with immobile GBM (n = 11). The fluorescence intensity of the gallbladder (GB) was significantly higher in dogs with normal gallbladders than that of dogs with GBM. FI was higher in dogs with mobile GBM than in immobile GBM dogs. Columns indicate mean values, and vertical error bars represent standard deviations. *P<0.001 between groups.

devoid of radiation exposure. It is a convenient procedure that does not require space-occupying devices such as a C-arm or assistance in performing the device [17].

The significance of IOC in veterinary medicine has been acknowledged [23–25]. While bile duct flushing during cholecystectomy removes mucus and inflammatory debris from the CBD [26], it may not perfectly rule out the possibility of partial obstruction, stricture, or other debris [23]. In contrast, cholangiography offers a precise assessment of the patency of the CBD and concurrently reduces the risk of bile duct injury [27]. In one study, CT cholangiography was reported in dogs, underscoring the necessity of this imaging system for assessing EHBO [24].

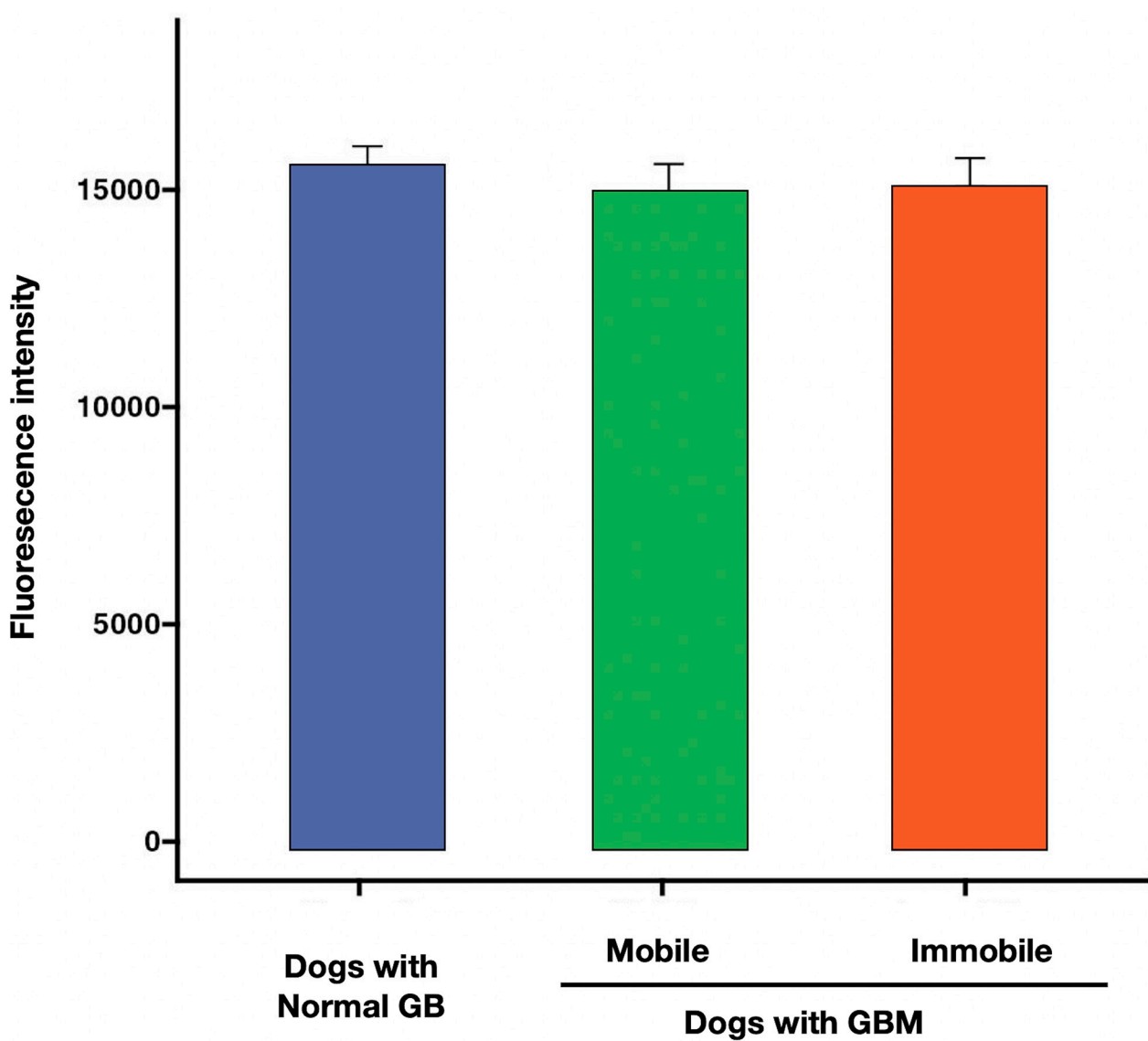

**Fig 6. Fluorescent intensity of the common bile duct (CBD).** The fluorescence intensity of the common bile duct did not significantly differ between dogs with a normal gallbladder (GB), mobile GBM, or immobile GBM. Comparison of fluorescence intensity of the CBD among the following groups: Controls with normal GB (n = 10), dogs with mobile GBM (n = 6), and dogs with immobile GBM (n = 11). Columns indicate mean values, and vertical error bars represent standard deviations. *P<0.001 between groups.

Other studies have demonstrated IOC using the C-arm in the veterinary field, highlighting its advantages over CT in terms of accuracy and real-time imaging [23, 25]. However, radiological imaging of the biliary tract may only be used selectively because the process takes time, involves radiation exposure, and needs some additional equipment and manpower for the procedure [17]. Hence, in veterinary medicine, there is a need for NIR fluorescence cholangiography, which is promising for the facile intraoperative identification of biliary anatomy. A recent study favorably mentioned the implementation of ICG cholangiography in eight healthy dogs [9]. ICG cholangiography with an IV injection of ICG was successfully performed in all eight dogs, providing guidelines for NIRF cholangiography in clinically normal dogs.

Visualization of the biliary tract is crucial during cholecystectomy, particularly in canine patients with GBM, where confirmation of CBD patency is crucial [26]. Catheterization and

flushing of the CBD before cystic duct ligation have been proposed to reduce the risk of post-operative compromised bile flow from choleliths or residual bile plugs and to ensure CBD patency [3, 28, 29]. Nevertheless, the need for intraoperative catheterization during cholecystectomy is still controversial [4]. The routine flushing of the CBD in patients with GBM may not be beneficial and may even lead to associated morbidity. Retrograde catheterization necessitates duodenotomy to access the major duodenal papilla (MDP), which can lead to prolonged surgical duration, increased anesthesia time, and a higher risk of complications such as intestinal adhesions and perforations. Furthermore, this procedure may adversely affect gastrointestinal motility. Reported complications related to retrograde catheterization include gallbladder necrosis, hepatic abscessation, and duodenal dehiscence or perforation [3]. Acute pancreatitis carries a high mortality rate in veterinary medicine, emphasizing the clinical significance of this complication [30]. Previous studies have shown that CBD catheterization, regardless of the approach, can increase surgical time without significant postoperative bilirubin level improvement [31, 32]. This suggests that CBD flushing may lead to the reflux of biliary secretions into the pancreatic duct, potentially causing pancreatitis, or result in trauma to the major duodenal papilla, leading to inflammation and temporary occlusion of the pancreatic duct orifice. Given these risks, it is crucial to consider alternative techniques. In NIR fluorescence IOC, proposed in this study, can ensure the patency of the CBD without the risk of catheterization and flushing. In this study, all 27 dogs with confirmed CBD patency exhibited FI similar to that of the CBD. However, when comparing the CBD in the GB, the GBM group showed a lower FI of the GB than the normal group, indicating that the mucocele did not manifest under ICG. This indicates that NIR fluorescence IOC facilitates the visualization of the biliary tract and enables the detection of mucocele presence, thereby confirming the patency of the CBD. This approach of using IOC with an IV injection of ICG is a novel technique that offers a less invasive approach for assessing biliary tract patency and visualization and enhances postoperative outcomes.

In this study, ICG was administered intravenously at 0.25 mg/kg at least 45 minutes before the surgery. The timing and dosages of ICG injection were decided because they showed effective intraoperative identification of biliary anatomy practically and safely in a human study [33]. After IV injection, more than 95% of ICG was cleared by hepatocytes and excreted into the bile within 15 min [5]. The FI of CBD peaks within 2–8 h depending on the dosage and species, and then gradually decreases [8, 34]. In one study, doses ranging from 0.05 to 0.25 mg/kg were used for cholangiography in healthy dogs [9]. Ultimately, when comparing our results, the dosage of 0.25 mg/kg itself is considered appropriate for evaluating patency. However, since only healthy dogs were compared, additional research is needed to determine the results based on dosage depending on pathological conditions. Therefore, further studies are warranted to determine optimal dosages based on pathological conditions, especially in dogs with GBM. In this study, the FI analyzing software (MGViewer V1.1.1, MetapleBio Inc.) was used to measure the absolute FI of the CBD in different cases. Previous studies have reported that ICG administration is safe, and well tolerated [35, 36]. Approximately 0.003% of anaphylactic reactions occur at doses higher than 0.5 mg/kg [37]. Because 0.25 mg/kg was intravenously injected for cholangiography in this study, the risk was thought to be still relatively low. However, it is important to note that while the dogs in this study did not experience any side effects from ICG, further research is necessary, as there have been no safety studies or recommended dosages reported specifically for dogs.

This study had a few limitations. First, the sample size of GB mucocele patients was small, limiting the generalizability of the findings to a broader population. Therefore, a study design with a larger sample size is required to provide more precise and reliable results and confirm our findings. Second, there were some limitations in assessing the optimal timing and dosage

of ICG administration during fluorescent IOC, which could affect the accuracy and effectiveness of the technique. Further well-designed studies are warranted to address these limitations and provide more robust evidence on the optimal timing and dose of ICG administration during fluorescent intraoperative cholangiography.

## Conclusion

In conclusion, this study demonstrates the application of NIR fluorescence IOC with IV injection of ICG in dogs. The cholangiography with ICG effectively visualized the biliary tract with minimal invasiveness, allowing for a comprehensive assessment of the CBD patency and delineation of biliary tree structures during cholecystectomy. While the study highlights its potential in canine patients with GBM, it is important to acknowledge certain limitations, particularly related to the representativeness of the study population and the necessity of catheterization in cases of obstruction. Future studies with larger sample sizes and refined methodologies to explore the optimal timing and dosage of ICG administration are required to validate the findings of this study and to further elucidate the clinical utility of this innovative technique in veterinary medicine.

## Supporting information

**S1 Fig. 1 fluorescence intensity of gallbladder and common bile duct of all patients.** (PDF)

## Author Contributions

**Conceptualization:** Su-Hyeon Kim, Sungin Lee.

**Data curation:** Su-Hyeon Kim, Sungin Lee.

**Formal analysis:** Su-Hyeon Kim, Sungin Lee.

**Funding acquisition:** Sungin Lee.

**Investigation:** Su-Hyeon Kim.

**Methodology:** Su-Hyeon Kim.

**Supervision:** Sungin Lee.

**Visualization:** Su-Hyeon Kim.

**Writing – original draft:** Su-Hyeon Kim.

**Writing – review & editing:** Su-Hyeon Kim, Sungin Lee.

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
