## [Decision Letter · Decision Letter 0]

18 Jan 2024

PONE-D-23-36283Determining the patency of biliary tracts in dogs with gallbladder mucocele using near-infrared cholangiography with indocyanine greenPLOS ONE

Dear Dr. Lee,

Thank you for submitting your manuscript to PLOS ONE. After careful consideration, we feel that it has merit but does not fully meet PLOS ONE’s publication criteria as it currently stands. Therefore, we invite you to submit a revised version of the manuscript that addresses the points raised during the review process.

We look forward to receiving your revised manuscript.

Kind regards,

Wenguo Cui, Ph.D

Academic Editor

PLOS ONE

Journal Requirements:

"This work was supported by the National Research Foundation of Korea (NRF) grant funded by the Korea government (MSIT) (RS-2023-00253736) and also supported by the Basic Research Lab Program (2022R1A4A1025557) through the National Research Foundation (NRF) of Korea, funded by the Ministry of Science and ICT."

"This work was supported by the National Research Foundation of Korea (NRF) grant funded by the Korea government (MSIT) (RS-2023-00253736) and also supported by the Basic Research Lab Program (2022R1A4A1025557) through the National Research Foundation (NRF) of Korea, funded by the Ministry of Science and ICT."

"This work was supported by the National Research Foundation of Korea (NRF) grant funded by the Korea government (MSIT) (RS-2023-00253736) and also supported by the Basic Research Lab Program (2022R1A4A1025557) through the National Research Foundation (NRF) of Korea, funded by the Ministry of Science and ICT."

5. Please provide a complete Data Availability Statement in the submission form, ensuring you include all necessary access information or a reason for why you are unable to make your data freely accessible. If your research concerns only data provided within your submission, please write "All data are in the manuscript and/or supporting information files" as your Data Availability Statement.

6. We note that Figure 2 in your submission contain copyrighted images. All PLOS content is published under the Creative Commons Attribution License (CC BY 4.0), which means that the manuscript, images, and Supporting Information files will be freely available online, and any third party is permitted to access, download, copy, distribute, and use these materials in any way, even commercially, with proper attribution. For more information, see our copyright guidelines: http://journals.plos.org/plosone/s/licenses-and-copyright.

Reviewers' comments:

Reviewer's Responses to Questions

**Comments to the Author**

1. Is the manuscript technically sound, and do the data support the conclusions?

Reviewer #1: Partly

Reviewer #2: Yes

2. Has the statistical analysis been performed appropriately and rigorously? 

Reviewer #1: Yes

Reviewer #2: Yes

3. Have the authors made all data underlying the findings in their manuscript fully available?

Reviewer #1: No

Reviewer #2: Yes

4. Is the manuscript presented in an intelligible fashion and written in standard English?

Reviewer #1: Yes

Reviewer #2: Yes

5. Review Comments to the Author

Reviewer #1: Overview:

This study describes and evaluates an interesting and innovative technic that would be of great help in hepatobiliary surgery.

In general, the manuscript is well written, concise and readable. Some revision would be interesting to

improve description of the material and methods and the results and to make the discussion clearer.

However, a rather major issue in the manuscript is related to the cases included in the GBM group compared the more general statements made from the data. The inclusion criteria and so the study population are rather well described and correspond to some cases of GBM seen in clinical practice; therefore, most conclusions are valid. However, the study population does not represent all cases of GBM. This recruitment bias should be more clearly acknowledged and represent a restraint to some claims, especially the evaluation of biliary ducts patency and alleviation for catheterization. All these limitations should be highlighted and the corresponding statements should be mitigated by adding on which cases they are applicable.

First, the study presents enough data to conclude that this technique permits to visualize the biliary tract in healthy dogs and dogs with GBM but without complete duct obstruction. However, it would also be nice to discuss whether a complete obstruction could limit the visualization of the biliary tract, as suggested from the images obtained from the dog presented in figure 3C. As this situation was not evaluated in this study and in the pilot one in dogs (reference 26), it would be nice to give data from the human literature.

As ICG was observed in the duodenum in all dogs, at least partial patency can be considered and therefore demonstrated with this technic. However, it should be commented and discussed as to whether partial obstruction was still possible. I think that cases with obstruction should have been included to definitively conclude on the use of ICG for the assessment of the patency of the cystic and common bile duct. While it seems judicious to hypothesize (as done lines 243-244) that it could permit to demonstrate obstruction, the data presented here cannot permit to conclude definitively.

The authors stated that this technic can eliminate the need for surgical catherization and flushing. It seems to be an over statement and so need to be mitigated. While it is probably true when the ducts seem patent, catherization would still be required in case of obstruction to remove the obstacle.

Specific comment

Concerning question 3 (Have the authors made all data underlying the findings in their manuscript fully available), I answered no as I understood that the data policy requires that individual data are available such as the fluorescence intensity for each dog (and not a median for the group), etc. If confirmed by the Editors that it is indeed required, I would suggest to give these data as a supporting information.

« The PLOS Data policy requires authors to make all data underlying the findings described in their manuscript fully available without restriction, with rare exception (please refer to the Data Availability Statement in the manuscript PDF file). The data should be provided as part of the manuscript or its supporting information, or deposited to a public repository. For example, in addition to summary statistics, the data points behind means, medians and variance measures should be available. »

Abstract:

The abstract is concise and captures some essential features of the manuscript but should be rewritten to be more accurate and to highlight the most relevant information. For example, it should precise that for GBM cases, only elective procedures were included; how fluorescence was assed and quantified, what were the criteria to define bile duct patency. On the contrary, some information is not essential such as “that underwent preoperative assessments, including imaging and biochemical analyses”. Concerning the results, beyond CBD patency and gallbladder intensity, adequate visualization of the anatomy of the remaining extrahepatic biliary tree could be mentioned.

In the view of the limits discussed previously, the conclusion seems overstated and should be mitigated.

Line 30: if only one study has indeed been published (reference 26), “only a few studies” should be modified for “only one”

Line 37: please precise between what significant difference were observed or consider removing the sentence if this information is not crucial.

Introduction

The introduction is appropriate, clearly explain why the subject matters and provide some rational for the study. However, the “few studies” (lines 75-76) should be referenced and commented to justify why the present study was needed.

Materials and methods

This section could be improved by defining more precisely some criterions.

The study seems prospective but it should be clearly stated and the time period for recruitment (line 113) could be mentioned elsewhere than it the “informed consent” paragraph. How the number of cases to be included was determined? It might be more accurate to describe the recruitment criteria in the material and methods section but only give the number of animals effectively included in the results section, except if it was pre-determined.

The criteria to diagnose GBM in general and to differentiate mobile vs immobile case should be define in this section (and not in the discussion). The criteria to define bile duct patency should also be clearly stated.

Could the authors please indicate which side effects were potentially expected and how were they monitored?

Line 119: it would be better to precise also the maximum time period between the injection and the start of the surgery

Statistical analysis: Lines 147-148 it is stated that data were presented as median but it seems that for figures 4 and 5, mean and standard deviation were used. Please, complete the statistical analysis accordingly.

Results:

While the results section already give relevant information, it could be partly rewritten to strengthen the data.

Lines 183-198: this part could benefit from some rewriting to improve the understanding. While information about the observation of all parts the biliary tree is given for the control, it is not the case for the GBM group. Lines 191-192 it is not clear whether the statement about the cystic duct concerned the entire group or the immobile GBM cases. Lines 195-196: The order of the sentences might also be improved, for example data about mucoceles are given lines 188 to 191 but again at line 194; data about CBD/duodenum lines 192-193 but again line 197. Lines 194-196, it might be more clearly stated whether any flushing was performed and if no flushing was performed why.

Line 197. If available, it could be interesting to add data regarding the length of the surgeries in each group to give information about how long the fluorescence was observed and also to be capable to comment in the discussion on whether this technic could shorten the surgical time.

Did any clinical, ultrasonographic and biological data are available about the short and long-term outcomes of the GBM cases? This information would be off great value to strengthen the statement that no side-effect from ICG developed and that no case had CBD obstruction (mainly partial obstruction) that had been missed during surgery

Table 1: maybe change “female” and “male” for “entire female” and “entire male” as usually “entire” is added to differentiate between the general gender and the reproductive status.

Figure 3: it seems that yellow and not green arrow heads can be seen on figure 3A. “HD” on figure 3B is not explicated in the legends.

Adding an image of “patency” with ICG within the duodenum would be nice.

Discussion:

In general, this section points out interesting and relevant data but it is a bit difficult to follow. In light to the previous comments, some section of the discussion would require modification, especially regarding the limitations.

The sequencing is sometimes confusing or not logical or the relevance of the information regarding the study questionable. Especially in the paragraph lines 245 to 261, the general information regarding GBM is not highly relevant and does not really explain why only elective procedure was considered.

The complications related to catherization are first mentioned line 262-263 but also again lines 303. It appears a bit redundant, while each time they are not clearly described. To better justify the search for alternative technics, a more thorough description of the complication related to catheterization would be nice.

Lines 297-300: it seems that this reference 26 is the only previous one using ICG cholangiography in dogs and it should be discussed more. Observation obtained in healthy dog in reference 26 could be compared to the observation in the current study in the control group. While references to protocols used in humans are made starting line 323, the authors could precise whether the materials and methods they used for ICG cholangiography was identical or different to the methods used in reference 26 and, to discuss why they eventually deviate from the published protocol in dogs.

Lines 318-320: this sentence seems confusing, please rephrase.

Lines 330-332: the sentence is also confusing as “the present study” would refer to the study described in the manuscript (that indeed report data regarding safety but not “less likely to cause side effects”) a

Conclusion

In light to previous comments, the conclusion should be rewritten taking into considering what was indeed demonstrated in the study and onto which cases the observation could be applied.

Lines 349-351: please give context and rephrase or remove the sentence as it is difficult to understand.

Hoping to read a 2nd version soon.

Reviewer #2: The manuscript entitled “Determining the patency of biliary tracts in dogs with gallbladder mucocele using near-infrared cholangiography with indocyanine green” introduced a study aiming to use ICG for IOC to identify fluorescent biliary tract images and determine the patency of the common bile duct during cholecystectomy in dogs. The study demonstrated that fluorescent IOC administration via intravenous injection of ICG can eliminate the need for surgical catheterization and flushing of the biliary ducts, which allows the assessment of the patency of the cystic duct and common bile duct during cholecystectomy.

There are some questions worth considering and addressing, so major revisions should be conducted before further consideration.

1. The full name of CBD should be supplemented in the abstract.

2. Tolerance of humans and animals to the same drug is very different. Generally speaking, the drug tolerance of animals is larger than people, which means animals need higher drug doses than humans. So, has the author adjusted the drug dose (0.25mg/kg was used in the study)?

3. Where is the scale bar of Figure 3?

4. Whether the authors measure the residual amount of ICG in other organs, such as the kidney, heart, spleen, and lung?

5. Is there any difference in the results between different dog types, especially between large and small dogs?

6. This research only studied whether IOC can be applied in canine cholangiography, there are limited experiments and data. I suggest more investigation into the dose of ICG administration and the detection time after ICG administration because these supplements in the experiment improve the scientific significance and reference value of this article.

7. Has the author tested the feasibility of this method on other animals?

8. There is a clerical error on Page 5 line 121 “0.25/mL”.

9. The author should pay attention to the space between numbers, symbols, and letters in the figure legends. For example, “n=6”, and “p=0.002” in page 7.

6. PLOS authors have the option to publish the peer review history of their article (what does this mean?). If published, this will include your full peer review and any attached files.

Reviewer #1: No

Reviewer #2: No

---

## [Author Response · Author response to Decision Letter 0]

17 Feb 2024

[Reviewer #1] 

* Overview

However, a rather major issue in the manuscript is related to the cases included in the GBM group compared the more general statements made from the data. The inclusion criteria and so the study population are rather well described and correspond to some cases of GBM seen in clinical practice; therefore, most conclusions are valid. However, the study population does not represent all cases of GBM. This recruitment bias should be more clearly acknowledged and represent a restraint to some claims, especially the evaluation of biliary ducts patency and alleviation for catheterization. All these limitations should be highlighted and the corresponding statements should be mitigated by adding on which cases they are applicable. First, the study presents enough data to conclude that this technique permits to visualize the biliary tract in healthy dogs and dogs with GBM but without complete duct obstruction. However, it would also be nice to discuss whether a complete obstruction could limit the visualization of the biliary tract, as suggested from the images obtained from the dog presented in figure 3C. As this situation was not evaluated in this study and in the pilot one in dogs (reference 26), it would be nice to give data from the human literature. As ICG was observed in the duodenum in all dogs, at least partial patency can be considered and therefore demonstrated with this technic. However, it should be commented and discussed as to whether partial obstruction was still possible. I think that cases with obstruction should have been included to definitively conclude on the use of ICG for the assessment of the patency of the cystic and common bile duct. While it seems judicious to hypothesize (as done lines 243-244) that it could permit to demonstrate obstruction, the data presented here cannot permit to conclude definitively. The authors stated that this technic can eliminate the need for surgical catherization and flushing. It seems to be an over statement and so need to be mitigated. While it is probably true when the ducts seem patent, catherization would still be required in case of obstruction to remove the obstacle.

Response: We sincerely appreciate the thorough review conducted by the reviewer and their valuable suggestions for enhancing our manuscript. Their insights have significantly contributed to improving the quality of our work. We have revised the manuscript as per the reviewer’s comments and believe that these revisions have considerably improved the manuscript. We have addressed each of your concerns individually in the responses below.

* Specific Comment

1. Concerning question 3 (Have the authors made all data underlying the findings in their manuscript fully available), I answered no as I understood that the data policy requires that individual data are available such as the fluorescence intensity for each dog (and not a median for the group), etc. If confirmed by the Editors that it is indeed required, I would suggest to give these data as a supporting information. « The PLOS Data policy requires authors to make all data underlying the findings described in their manuscript fully available without restriction, with rare exception (please refer to the Data Availability Statement in the manuscript PDF file). The data should be provided as part of the manuscript or its supporting information, or deposited to a public repository. For example, in addition to summary statistics, the data points behind means, medians and variance measures should be available. »

Response: We appreciate your valuable suggestion. In response to the reviewer's suggestion, we have included the necessary data as supporting information to ensure compliance with the PLOS Data Policy. Thank you for guiding us through this process.

* Abstract

2. The abstract is concise and captures some essential features of the manuscript but should be rewritten to be more accurate and to highlight the most relevant information. For example, it should precise that for GBM cases, only elective procedures were included; how fluorescence was assessed and quantified, what were the criteria to define bile duct patency. 

Response: We thank the reviewer for the pertinent comments and suggestions. First, we additionally mentioned the elective procedure (page 2, lines 33-34 in the revised manuscript). Furthermore, the paragraph was rewritten, and a more detailed description was introduced, to improve clarity (Page 2, lines 35-39 in the revised manuscript).

3. On the contrary, some information is not essential such as “that underwent preoperative assessments, including imaging and biochemical analyses”. 

Response: Thank you for your comment. We deleted this part.

4. Concerning the results, beyond CBD patency and gallbladder intensity, adequate visualization of the anatomy of the remaining extrahepatic biliary tree could be mentioned. 

Response: We thank the reviewer for this valuable suggestion. As suggested, we additionally mentioned the visualization of the extrahepatic biliary tree (Page 2, lines 41-42 in the revised manuscript).

5. In the view of the limits discussed previously, the conclusion seems overstated and should be mitigated.

Response: Thank you for your valuable comment. As suggested, we rephrased the conclusion to better align with the acknowledged limitations discussed earlier (Page 2, lines 42-47 in the revised manuscript).

6. Line 30: if only one study has indeed been published (reference 26), “only a few studies” should be modified for “only one”

Response: We revised this sentence according to your suggestion (page 2, line 30 in the revised manuscript).

7. Line 37: please precise between what significant difference were observed or consider removing the sentence if this information is not crucial.

Response: We appreciate the suggestion. To improve clarity, we deleted this sentence to improve clarity.

* Introduction

8. The introduction is appropriate, clearly explain why the subject matters and provide some rational for the study. However, the “few studies” (lines 75-76) should be referenced and commented to justify why the present study was needed.

Response: We sincerely appreciate valuable suggestions for improvement. First, we have changed “few studies” to “only one study” and added the reference. Second, this paragraph was further rewritten, and a more detailed description was introduced, to justify why the present study was needed (page 4, lines 76-78 in the revised manuscript).

* Materials and methods 

This section could be improved by defining more precisely some criteria.

9. The study seems prospective but it should be clearly stated and the time period for recruitment (line 113) could be mentioned elsewhere than it the “informed consent” paragraph. 

Response: We appreciate this suggestion. As per the reviewer’s suggestion, we deleted about time period in the “informed consent” and inserted it into the “study population and preoperative assessment” section (page 4, lines 87-88 in the revised manuscript).

10. How the number of cases to be included was determined? It might be more accurate to describe the recruitment criteria in the material and methods section but only give the number of animals effectively included in the results section, except if it was pre-determined.

Response: We thank you for your inquiry and valuable suggestions. First, for the determination of the number of cases, we set a specific period (Client-owned dogs were prospectively recruited for the study from February 2021 until August 2023.) and determined the number of cases by categorizing them based on the patients collected within that period. Second, as you suggested, we removed some sentences and gave only the number of animals effectively included in the results section and rewritten in the material and methods section to improve readability (page 5, lines 100-102 in the revised manuscript).

11. The criteria to diagnose GBM in general and to differentiate mobile vs immobile case should be define in this section (and not in the discussion). 

Response: We appreciate your suggestion. As suggested, the criteria to diagnose GBM was removed from the Discussion section and included in the material and methods section (Page 5, lines 102-109 in the revised manuscript). 

12. The criteria to define bile duct patency should also be clearly stated.

Response: We appreciate the insightful comment, and this clarification enhances the understanding of our methodology and criteria for assessing bile duct patency. The criteria to define bile duct patency were additionally discussed (Pages 6-7, lines 144-148 in the revised manuscript).

13. Could the authors please indicate which side effects were potentially expected and how were they monitored?

Response: We are very thankful for the valuable suggestion which has enhanced the quality of the research. We have created a new section regarding monitoring and provided additional explanations (Page 7, lines 154-163 in the revised manuscript).

14. Line 119: it would be better to precise also the maximum time period between the injection and the start of the surgery

Response: We appreciate this suggestion. As per the reviewer’s suggestion, this sentence was revised (Page 6, lines 129-130 in the revised manuscript).

15. Statistical analysis: Lines 147-148 it is stated that data were presented as median but it seems that for figures 4 and 5, mean and standard deviation were used. Please, complete the statistical analysis accordingly.

Response: We thank the reviewer for pointing this out, and we apologize for this error. We determined that the manuscript description was wrong, and therefore, we have changed “Data were presented as the median value for each group, followed by the interquartile range” to “Data were presented as means and standard deviations.” (Page 8, lines 172-173 in the revised manuscript).

* Results

16. Lines 183-198: this part could benefit from some rewriting to improve the understanding. While information about the observation of all parts the biliary tree is given for the control, it is not the case for the GBM group.

Response: We thank the reviewer for the suggestion. We have introduced the necessary changes in the revised manuscript (page 10, lines 215-218).

17. Lines 191-192 it is not clear whether the statement about the cystic duct concerned the entire group or the immobile GBM cases.

Response: We apologize for the confusion. We have modified the sentence to make it clearer (Page 10, lines 217-218 in the revised manuscript).

18. Lines 195-196: The order of the sentences might also be improved, for example data about mucoceles are given lines 188 to 191 but again at line 194; data about CBD/duodenum lines 192-193 but again line 197.

Response: Thank you for your valuable comments which helped to enhance readability. We have modified the sentence to make it clearer. (Page 10, lines 215-223 in the revised manuscript)

19. Lines 194-196, it might be more clearly stated whether any flushing was performed and if no flushing was performed why.

Response: We appreciate the reviewer for this feedback. We have revised the manuscript per your suggestion (Page 10, lines 218-220 in the revised manuscript). Additionally, to address the reason for not performing flushing, we have revised and added the following explanation the Methods & Materials section: "Bile duct patency was determined by comparing FI values. In the control group, values for the gallbladder (GB) and common bile duct (CBD) were established. When FI values of the CBD in GBM patients closely resembled those in the control group, CBD patency was considered confirmed. This criterion ensured a thorough evaluation of bile duct patency while minimizing unnecessary interventions.” (Pages 6-7, lines 144-148 in the revised manuscript).

20. Line 197. If available, it could be interesting to add data regarding the length of the surgeries in each group to give information about how long the fluorescence was observed and also to be capable to comment in the discussion on whether this technique could shorten the surgical time.

Response: Thank you for your valuable suggestion. In this study, we focused solely on elective patients and primarily on GB cholangiography; therefore, we did not measure the duration of the surgeries this time. However, in patients with CBD obstruction, the process of flushing into the duodenum could be skipped if the patency is confirmed by ICG cholangiography. And, this could contribute significantly to shortening the surgical time. In further studies, we are currently investigating patients with obstructive conditions using ICG cholangiography, both with and without flushing, and comparing surgical times, to explore this aspect further.

21. Did any clinical, ultrasonographic and biological data are available about the short and long-term outcomes of the GBM cases? This information would be off great value to strengthen the statement that no side-effect from ICG developed and that no case had CBD obstruction (mainly partial obstruction) that had been missed during surgery

Response: We appreciate your valuable comments. We have included additional information in the revised manuscript (Pages 10, lines 224-229 in the revised manuscript). We are pleased to report that all patients showed favorable outcomes, with no complications observed after surgery. Although individual patient data were considered for inclusion, we decided to prioritize the focus on GB cholangiography in our manuscript. Therefore, separate figures for radiographic or ultrasonographic data were not included.

22. Table 1: maybe change “female” and “male” for “entire female” and “entire male” as usually “entire” is added to differentiate between the general gender and the reproductive status.

Response: We thank the reviewer for this valuable suggestion. We revised this sentence as suggested (Page 9, line 196 in the revised manuscript).

23. Figure 3: it seems that yellow and not green arrow heads can be seen on figure 3A. “HD” on figure 3B is not explicated in the legends.

Response: We appreciate the reviewer for bringing this to our attention, and we apologize for the oversight. There was a mistake in the ordering of Figure 3 A and B. We have corrected this error by rearranging the figures to align with the correct sequence (Figure 3 on the re-uploaded file).

24. Adding an image of “patency” with ICG within the duodenum would be nice.

Response: We sincerely appreciate valuable suggestions for improvement. We have added a new figure (Figure 4) and figure legend that describes the “patency” with ICG within the duodenum (Page 11, lines 140-142 in the revised manuscript).

* Discussion

25. The sequencing is sometimes confusing or not logical or the relevance of the information regarding the study questionable. Especially in the paragraph lines 245 to 261, the general information regarding GBM is not highly relevant and does not really explain why only elective procedure was considered.

Response: We apologize for the confusion and appreciate the reviewer's suggestion which enhanced the clarity and readability of the manuscript. 

We have modified the paragraph (Page 13, lines 279-291 in the revised manuscript). Also, we relocated some content to the materials and methods section and adjusted the sentences accordingly (Page 5, lines 100-109 in the revised manuscript). We included general information about GBM to provide context for the comparison of blood tests among the groups in the results section, as there were notable differences observed (Page 13, lines 279-282 in the revised manuscript).

Second, the decision to focus solely on elective surgeries was made because this study represents an initial step in our research journey, we aimed to establish a foundation of credibility and reliability. However, we acknowledge the importance of further investigations involving patients with CBD obstruction, which are currently in the planning stages.

26. The complications related to catheterization are first mentioned line 262-263 but also again lines 303. It appears a bit redundant, while each time they are not clearly described. To better justify the search for alternative technics, a more thorough description of the complication related to catheterization would be nice.

Response: We thank the reviewer for your valuable suggestions. We have addressed the issue of redundancy by revising the manuscript to eliminate duplicated content (Page 13, lines 288-289 in the revised manuscript). Additionally, we have added further details regarding complications related to catheterization (Page 15, lines 333-339 in the revised manuscript).

27. Lines 297-300: it seems that this reference 26 is the only previous one using ICG cholangiography in dogs and it should be discussed more. Observation obtained in healthy dog in reference 26 could be compared to the observation in the current study in the control group. While references to protocols used in humans are made starting line 323, the authors could precise whether the materials and methods they used for ICG cholangiography was identical or different to the methods used in reference 26 and, to discuss why they eventually deviate from the published protocol in dogs.

Response: We appreciate the reviewer’s valuable comments, and we acknowledge the importance of discussing Reference 26 more extensively in our manuscript. First, we would like to mention that our dosage determination was not solely based on Reference 26, as our experiments were conducted before the publication of this reference. Instead, we initially determined the dosage based on references in human medicine. Second, it should be noted that our study focused on evaluating the efficacy of a single dosage rather than comparing outcomes across various dosages. Reference 26 utilized dosages ranging from 0.05 to 0.25 mg/kg, whereas we specifically employed a dosage of 0.25 mg/kg. However, upon comparing our results with those of Reference 26, we confirmed that the dosage of 0.25 mg/kg proved to be effective in assessing patency. Also, it is crucial to emphasize that further research is warranted to explore the effects of varying dosages based on pathological conditions. We have discussed these points in the discussion section. (Page 16, lines 360-366 in the revised manuscript).

28. Lines 318-320: this sentence seems confusing, please rephrase.

Response: Thank you for your suggestion. We have revised the sentence to improve clarity (Page 16, lines 350-352 in the revised manuscript).

29. Lines 330-332: the sentence is also confusing as “the present study” would refer to the study described in the manuscript (that indeed report data regarding safety but not “less likely to cause side effects”) 

Response: We apologize for the confusion and appreciate your comment. We have revised the paragraph to ensure clarity (Page 16, lines 367-373 in the revised manuscript).

* Conclusion

30. In light to previous comments, the conclusion should be rewritten taking into considering what was indeed demonstrated in the study and onto which cases the observation could be applied.

Response: We thank the reviewer for this valuable comments. We have revised the conclusion as per the reviewer’s comments and believe that these revisions have considerably improved the conclusion (Page 17, lines 384-393 in the revised manuscript).

31. Lines 349-351: please give context and rephrase or remove the sentence as it is difficult to understand.

Response: We appreciate this suggestion. As per the reviewer’s suggestion, we deleted this sentence to improve clarity.

[Reviewer #2] 

* Major revisions

1. The full name of CBD should be supplemented in the abstract.

Response: We thank the reviewer for this suggestion, and we introduced the necessary changes (page 2, line 38 in the revised manuscript).

2. Tolerance of humans and animals to the same drug is very different. Generally speaking, the drug tolerance of animals is larger than people, which means animals need higher drug doses than humans. So, has the author adjusted the drug dose (0.25mg/kg was used in the study)?

Response: We appreciate your valuable suggestion. As stated in the discussion (lines 355-360), we determined the final dosage based on various references from human medicine. Specifically, one study referenced in our discussion (Reference 26) utilized dosages ranging from 0.05 to 0.25 mg/kg for cholangiography in healthy dogs. Upon comparing our results with this study, the dosage of 0.25 mg/kg appeared suitable for evaluating patency in both studies (Page 16, lines 360-366 in the revised manuscript). However, considering pathological conditions and other factors, further investigation is warranted to determine the optimal dose for dogs.

3. Where is the scale bar of Figure 3?

Response: We appreciate your valuable suggestion. Unfortunately, the imaging device used in our study does not provide a scale reference for real-time visualization of organ sizes. While we did measure the size of the excised gallbladders post-surgery, comparing them to the intraoperative images may introduce inaccuracies due to perspective distortion. Therefore, we could not include a scale bar in Figure 3 to avoid potential inaccuracies in size estimation.

4. Whether the authors measure the residual amount of ICG in other organs, such as the kidney, heart, spleen, and lung?

Response: Thank you for the reviewer’s comment. In this study, our primary objective was to confirm the patency of the biliary tract and assess biliary tree visualization using cholangiography rather than measuring the residual amount of ICG in other organs. As such, our focus was directed towards the specific aims of this investigation. However, we acknowledge the importance of exploring the residual amount of ICG in other organs as a potential avenue for further research. Future studies could indeed delve into this aspect to provide a more comprehensive understanding of the pharmacokinetics and biodistribution of ICG in various tissues.

5. Is there any difference in the results between different dog types, especially between large and small dogs?

Response: We appreciate your inquiry. Upon reviewing the results of our study, we found minimal differences between different dog types, as indicated in Table 1. It was challenging to draw conclusive comparisons based solely on breed differences. Further investigation into potential variations between dog types, particularly between large and small breeds, may be warranted to provide a more comprehensive understanding of the implications of breed-specific factors on the outcomes of NIR cholangiography.

6. This research only studied whether IOC can be applied in canine cholangiography, there are limited experiments and data. I suggest more investigation into the dose of ICG administration and the detection time after ICG administration because these supplements in the experiment improve the scientific significance and reference value of this article.

Response: We appreciate your insightful suggestions. We are fully aware of the importance of further investigation into the dose of ICG administration and the detection time after ICG administration. These aspects are currently under consideration, and we are planning to incorporate them into our ongoing research efforts. Your suggestions will undoubtedly contribute to the refinement and advancement of our study, and we are committed to addressing them in our future work.

7. Has the author tested the feasibility of this method on other animals?

Response: We appreciate your valuable comment. In this study, we did not conduct feasibility testing on other animals. Instead, we relied on references from previous studies involving humans, or dogs to establish the method's safety and feasibility (see references below). Considering the established safety based on these references, additional experiments on other animal species were not pursued in this study. Nonetheless, we closely monitored the canine patients both immediately after injection and post-surgery, and no adverse reactions related to ICG were observed during this study.

References:

1. Zarrinpar A, Dutson EP, Mobley C, Busuttil RW, Lewis CE, Tillou A, et al. Intraoperative Laparoscopic Near-Infrared Fluorescence Cholangiography to Facilitate Anatomical Identification: When to Give Indocyanine Green and How Much. Surg Innov. 2016;23(4): 360-5.

2. HOPE-ROSS, Monique, et al. Adverse reactions due to indocyanine green. Ophthalmology, 1994;101(3): 529-533.

3. Larose PC, Brisson BA, Sanchez A, Monteith G, Singh A, Zhang M. Near-infrared fluorescence cholangiography in dogs: A pilot study. Vet Surg. 2023. 

4. GENÉ ŠKRABEC, Clara, et al. Fluorescent cholangiography with direct injection of indocyanine green (ICG) into the gallbladder: a safety method to outline biliary anatomy. Langenbeck's Archives of Surgery, 2020;405: 827-832.

8. There is a clerical error on Page 5 line 121 “0.25/mL”.

Response: We thank the reviewer for pointing this out, and we apologize for this error. We revised this sentence (page 5, line 121 in the revised manuscript).

9. The author should pay attention to the space between numbers, symbols, and letters in the figure legends. For example, “n=6”, and “p=0.002” in page 7.

Response: We appreciate the useful comments and criticisms provided by the reviewer, and, following a long and careful discussion, we agree with all the reviewer’s suggestions, and have introduced the appropriate changes in the remainder of the manuscript.

---

## [Editor Report · Decision Letter 1]

27 Feb 2024

Determining the patency of biliary tracts in dogs with gallbladder mucocele using near-infrared cholangiography with indocyanine green

PONE-D-23-36283R1

Dear Dr. Sungin Lee,

We’re pleased to inform you that your manuscript has been judged scientifically suitable for publication and will be formally accepted for publication once it meets all outstanding technical requirements.

Kind regards,

Wenguo Cui, Ph.D

Academic Editor

PLOS ONE

---

## [Editor Report · Acceptance letter]

21 Mar 2024

PONE-D-23-36283R1 

PLOS ONE

Dear Dr. Lee, 

I'm pleased to inform you that your manuscript has been deemed suitable for publication in PLOS ONE. Congratulations! Your manuscript is now being handed over to our production team.

Kind regards, 

on behalf of

Professor Wenguo Cui 

Academic Editor

PLOS ONE